# Carbon-Assistant Nanoporous Gold for Surface-Enhanced Raman Scattering

**DOI:** 10.3390/nano12091455

**Published:** 2022-04-25

**Authors:** Zhiyu Jing, Ling Zhang, Xiaofei Xu, Shengli Zhu, Heping Zeng

**Affiliations:** 1School of Optical-Electrical and Computer Engineering, University of Shanghai for Science and Technology, Shanghai 200093, China; 201310032@st.usst.edu.cn (Z.J.); 17625202808@163.com (X.X.); 2School of Materials Science and Engineering, Tianjin University, Tianjin 300350, China; slzhu@tju.edu.cn; 3State Key Laboratory of Precision Spectroscopy, East China Normal University, Shanghai 200241, China; hpzeng@phy.ecnu.edu.cn

**Keywords:** surface-enhanced Raman scattering, carbon assistance, crystal violet, finite difference time domain, enhancement factor, nanoporous gold

## Abstract

Surface-enhanced Raman scattering (SERS) technology can amplify the Raman signal due to excited localized surface plasmon (LSP) from SERS substrates, and the properties of the substrate play a decisive role for SERS sensing. Several methods have been developed to improve the performance of the substrate by surface modification. Here, we reported a surface modification method to construct carbon-coated nanoporous gold (C@NPG) SERS substrate. With surface carbon-assistant, the SERS ability of nanoporous gold (NPG) seriously improved, and the detection limit of the dye molecule (crystal violet) can reach 10^−13^ M. Additionally, the existence of carbon can avoid the deformation of the adsorbed molecule caused by direct contact with the NPG. The method that was used to improve the SERS ability of the NPG can be expanded to other metal structures, which is a convenient way to approach a high-performance SERS substrate.

## 1. Introduction

Surface-enhanced Raman scattering (SERS) is popularly employed in the fields of chemical analysis, medical diagnosis, food security [1,2] and sensing [3,4] due to its non-destructive nature and single-molecule-level sensitivity [5]. A number of different SERS substrates have been developed, including precious metals and non-precious metals, to achieve sensitive detection in different situations.SERS substrates based on non-precious metals, such as Ti, Zn, Mo and W, have been widely studied and applied because of their dominant chemical enhancement [6]. Xue et al. [7] found that TiO_2_ nanoparticles with a 10.9 nm diameter obtained 3.5 × 10^3^-fold enhancement. Wang et al. [8] prepared the amorphous ZnO nanocages and realized 6.62 × 10^5^-fold enhancement. However, the SERS stability and enhancement effect of non-noble metals is weaker than that of noble metals. Precious metals (Au, Ag, Cu) are still the best choice for preparing the SERS substrate, and researchers fabricate a variety of substrates with different nanostructures, such as a nanoporous structure [9], nanoparticle array [10], nanogap [11], etc. Nanoporous metals obtained by dealloying can possess a large number of directional or random pores and ligaments, fractures and other special structures, showing good SERS performance. It is generally believed that nanoporous metals, which have high specific surface area and three-dimensional interconnected channels, can concentrate the incident light and limit the surrounding light in the vicinity of the nanoscale pores, sharp corners and tip structures. Concomitantly, a strong local electromagnetic field will generate on nanoporous metal, resulting in excellent SERS ability [9,12]. Hu et al. [13] prepared nanoporous gold (NPG) via sputtering followed by dealloying, and the SERS enhancement factor was around 2.4 × 10^5^. Huang et al. [12] designed an NPG structure that contained a gradient of ligaments and pore structures along the thickness direction, and the SERS enhancement factor reached 10^7^. The wrinkled NPG films prepared by Zhang et al. [14] that contain rich broken ligaments with nanotips and nanogaps in between can produce a 0.7 × 10^8^ enhancement factor. Cai et al. [15] obtained graded NPG by two-step dealloying of AuAl alloy and combined macropores and nanopores to provide high-density hot spots and a larger specific surface area, showing an enhancement factor of 2.16 × 10^7^. Zhan et al. [16] reported double-layer nanoporous silver (NPS) prepared by dealloying with AgCu alloy and obtained an enhancement factor of 1.8 × 10^7^. By adjusting the pore/ligament size and porosity of each layer, more light can be captured due to the antireflection effect, leading to stronger SERS enhancement. Thus, it is possible to improve the SERS performance of nanoporous metals by constructing a hierarchical structure and surface modulation.

Carbon materials, such as graphene, graphene oxide (GO) and so on, have been used for SERS substrates due to their biocompatibility, low cost, nontoxic nature and stabilization. Since carbon material is a good candidate for coating modification [17,18,19,20,21,22], carbon-coated noble metal that combines the best of both possibly shows better SERS ability. Here, we report a chemical etching method to fabricate NPG with a carbon coating by pre-annealing Au_36_Cu_64_ (at. %) [23,24,25,26,27,28] ribbons covered with carbon materials at elevated temperature, and, after dealloying with diluted hydrochloric acid (HCl), carbon-assistant NPG (C@NPG) was formed. The carbon coating obviously improved the SERS performance of the NPG, and the detection limit of the C@NPG is about 10^−13^ M, with an average enhancement factor (EF) of 10^8^.

## 2. Experimental Methods

### 2.1. Preparation of Carbon-Assistant Nanoporous Gold

Figure 1 briefly summarizes the preparation process of carbon-assistant nanoporous gold. The alloy ribbons with a composition of Au_36_Cu_64_ (at.%) were prepared from Au (purity, 99.99 wt.%) and Cu (purity, 99.99 wt.%) by vacuum belt throwing furnace. The pre-alloyed materials were rapidly solidified into ribbons (thickness: ~30 μm; width: ~4 mm; length: several centimeters). Then, Au_36_Cu_64_ (at.%) ribbons were covered by carbon materials (weighing paper: 1 g) and annealed in the furnace at temperature of 300 °C for 12 h. After high-temperature oxidation, the annealed AuCu ribbons were soaked in HCl aqueous solution at room temperature for time-controlled dealloying. The corroded ribbons were washed several times with ultrapure water to remove residual hydrochloric acid, and then dried in the vacuum. The thickness of the final substrates is 20 μm (see Appendix A). In this process, copper compounds were dissolved preferentially as the more active component, leaving stable gold atoms to form nanoporous structure. Carbon element, which is stable at room temperature, remained and assembled on the surface of generated NPG.

### 2.2. Materials and Instruments

Copper and gold were supplied by Zhongnuo New Materials technology Company(Beijing, China). AuCu precursor alloy [Au_36_Cu_64_(at%)] was prepared by vacuum belt throwing furnace. Crystal violet (CV), nicotinamide (analytical standard, ≥99.8%) and 36.5% hydrochloric acid were purchased from Sinopharm Chemical Reagent Company (Shanghai, China).Weighing paper (butter paper) was used to serve as carbon element. Ultrapure water (18.2 MΩ) was manufactured by machine from Chengdu You Pu Biotechnology Co., Ltd (Ulupure, UPT, Chengdu, China). All Raman spectra were obtained using an RTS Raman spectrometer (Titan Electro-Optics Co., Ltd., Chengdu, China). The microstructure of the substrates was characterized by a scanning electron microscope (SEM, Quanta FEG 250, Thermo Fisher Scientific, Waltham, MA, USA) and a transmission electron microscope (TEM, Talos F200S, FEI Co., Ltd., Waltham, MA, USA). X-ray photoelectron spectroscopy (XPS) of the substrates was performed using an X-ray photoelectron spectrometer (AXIS, Shimadzu, Suzhou, China).

### 2.3. SERS Measurements

A drop of crystal violet (CV) aqueous solution (20 μL) was added onto the substrates for SERS detection. A 632.8 nm excitation laser with a low power of 1.0 mW was adopted for spectra measurement, and integral accumulation time of each Raman spectrum was 1 s (temperature 25 °C).

## 3. Results and Discussion

### 3.1. Morphology Characterization and SERS Ability Comparison of NPG with and without Carbon Material for Pretreatment after Dealloying

In order to explore the relationship between the structure and SERS properties, SERS spectra obtained from substrates prepared with different conditions were compared and the corresponding surface morphologies of the samples were characterized by SEM (Figure 2 and Appendix A). Figure 2a shows prepared NPG with AuCu alloy annealed at 200 °C (12 h) and corroded with 0.5 M hydrochloric acid for 24 h; the size of the pores and ligaments are identical to the diameter: ~40 nm. Figure 2b illustrates the AuCu alloy annealed at 300 °C (12 h) and corroded with 0.1 M hydrochloric acid for 30 min. In such a condition, discontinuous nanopores with the size around 100 nm distribute on the surface. Figure 2c shows the AuCu alloy with carbon materials annealed at 300 °C (12 h) and corroded with 0.1 M hydrochloric acid for 30 min. As shown in the figures, a bicontinous, nanoporous structure formed with the ligament and pore sizes around 200 nm to 600 nm, and the ligaments show a roughened surface. According to the Arrhenius formula, the diffusion coefficient is positively correlated with the absolute temperature. Therefore, the large pore sizes (Figure 2b,c) may be due to the stronger interdiffusion of Au and Cu at higher annealing temperatures. The melting and fusion processes produce the larger oxide grains, and gold atoms will occupy the interstices of copper oxide particles, resulting in the consequent increase in nanopore size after corrosion [26]. The addition of carbon materials during pre-annealing resulted in the formation of more continuous, uniform, hundreds-of-nanometers-long nanopores, and open ligament channel structures are formed on the surface after dealloying. The energy dispersive spectrometer (EDS) analyzing results (see Appendix A) show that, after dealloying, the mass ratio of gold and copper is about 0.56 before adding carbon material, and about 1.3 after adding carbon material. Obviously, in the presence of carbon materials, the mass ratio of gold and copper increased. According to the oxidation theory, the oxygen on the outer surface of the oxide layer is reduced to O^−^ and O^2−^, and the copper on the alloy surface will be oxidized to copper oxide and cuprous oxide. Secondly, carbon will react with oxygen at high temperatures to produce carbon dioxide. When the temperature drops, certain water vapor will be produced throughout the furnace cavity, which will also make part of the copper generate alkaline copper carbonate. Under the environment of dilute hydrochloric acid solution, the removal of copper compounds forms a better nanoporous structure with a rugged ligament. Thus, the existence of carbon can affect the oxidation and annealing process and is helpful for generating the porous structure.

The NPG with a coarse pore size of ~250 nm prepared by Kucheyev et al. [29] and the NPG with a nanopore size of 5 to 10 nm prepared with gold–silver alloy by Qian et al. [30] both can demonstrate an excellent reinforcement effect. The former one is due to the enhancement of the rough ligaments, and the latter is because of the local electromagnetic coupling effect. The pore size of the nanopore in Figure 2a belongs to the range of ultra-fine nanopore, which is conducive to the SERS enhancement. In Figure 2c, the pore size is large, up to hundreds of nanometers, but its ligaments are rough and with small tips on the surface, which can promote the intensity of the electromagnetic field and provide more active sites for molecular adsorption. From the TEM images shown in Figure 2d, it can be seen that a ~1-nm-thick amorphous carbon layer covers the ligament. Figure 2e compared the SERS spectra that collected on the NPG prepared with different conditions, and 633 nm laser with 1 mW power was adopted for the spectra measurement. The C@NPG substrate shows the better SERS enhancement (line III), which is ~40 times stronger than that of NPG without carbon (line II) and ~200 times stronger than NPG with smaller nanopores and ligaments prepared at a lower annealing temperature (line I). It indicates that, with the help of carbon material, it is possible to generate a more excellent nanostructured substrate for SERS applications.

### 3.2. Surface Carbon Analysis of Nanoporous Gold Pretreated with and without Carbon

The surface chemical states of the substrates with and without carbon during the annealing process (300 °C) were characterized by XPS. As shown in Figure 3, the peak positions of Cu (2p_2/3_ at 932.2 eV and 2p_1/2_ at 952 eV) and the peak positions of Au (4f_5/2_ at 84.3 eV and 4f_7/2_ at 87.9 eV) do not show any differences for the substrate with and without carbon. However, the relative intensity of the peaks associated between Cu and Au are the opposite, indicating that, with carbon-assistant, more copper element was removed from the surface during the dealloying process. The peak of Au 4f can be attributed to metallic Au, while the peaks of Cu 2p might belong to either metallic Cu or Cu oxide. In addition, the C (1s) peak at 284.8 eV, C-O peaks at 286 eV and 286.5 eV and carbon monoxide peak at 282.2 eV can be found in both samples, which might be caused by oxidation during XPS measurement. In fact, the annealing process is carried out in the atmosphere, and the existence of C 1s could be due to the absorption of gas [31,32]. However, the carbon states of the two samples are different. In particular, one peak represents O-C=O (289 eV), which may be derived from copper compounds existing in the sample without carbon, and the other new peak related to C-C/C graphite (284.1 eV) exists in the sample with carbon. It is speculated that the carbon source has carbonized in the heat treatment environment, and a small amount of C will be attached to the alloy surface through van der Waals force during the annealing process.

### 3.3. Mechanism Analysis of SERS Improvement

To understand the enhancement mechanisms of the enhanced SERS ability of C@NPG, the five characteristic Raman peaks of the totally symmetric and nontotally symmetric vibration of the CV molecule were investigated. Table 1 lists the five strongest characteristic Raman peaks from CV; the peaks at 442 cm^−1^ and 1624 cm^−1^ belong to the total symmetric vibration a1 modes, and the other three at 810 cm^−1^, 1177 cm^−1^ and 1379 cm^−1^ belong to the nontotally symmetric vibration e modes [15,33]. Comparing the position of the characteristic Raman peaks obtained from the NPG with and without carbon, four peaks, except the peak at 442 cm^−1^, exhibit a slight red-shift with carbon covering. Furthermore, the variation in the molecule on C@NPG is more significantly shifted compared to that of the free molecule in the solution and bare NPG, manifesting the carbon interlayer conduced to the absorption of the molecule onto the substrate. According to the Herzberg–Teller vibronic coupling mechanism, when surface plasmon resonance is the main contribution to the SERS enhancement, the intensities of the totally symmetric modes will be obviously improved; otherwise, the nontotally symmetric modes are more enhanced when charge transfer resonance dominates the SERS enhancement [33]. Both the total symmetric modes (e.g., 1624 cm^−1^) and nontotally symmetric modes (e.g., 810 cm^−1^) are comparably enhanced by about 40 times. Chemical enhancement is not dominant in C@NPG, while the electromagnetic enhancement plays a key role in the improved SERS performance.

The electromagnetic enhancement originating from a local surface photoelectric field mainly depends on the interaction between the light and metal surface. Thus, in order to distinctly clarify the effect of the carbon on the localized photoelectric field, the finite difference time domain method (FDTD) was adopted to simulate the electric field distribution. As shown in Appendix A, the EM field slightly attenuates with the increase in carbon thickness, but, when the thickness of the carbon layer is less than or equal to 1 nm, the area of the EM field with higher intensity obviously increased. The corresponding photoelectric field distributions of bare NPG and C@ NPG are displayed in Figure 4. It can be seen that, whether the carbon layer exists or not, strong local electromagnetic fields appear near the ligament surface, but the local electromagnetic field becomes stronger with the presence of the carbon layer. Taking |E/E_0_| (E is the local maximum electric field; E_0_ is the input source electric field in the linear simulation) as the scale, the maximum electric field enhancement coefficient is 15 for bare NPG (Figure 4a), while the maximum electric field enhancement coefficient with carbon reaches 23 (Figure 4b). Although the existence of the carbon layer only increases the field strength by a factor of nearly two, it can increase the SERS gain by a factor of 16 at the same point, and the space ratio of the strongest electromagnetic field is significantly higher, which can further improve the SERS capability of the substrate. Besides, the large π-bond system formed by the carbon molecular layer provides more possibilities for coordination bonds, and carbon has high stability and good biological affinity, which can reduce the interference of background fluorescence and improve the intensity of the Raman signal [34]. It is worth pointing out that the carbon layer, as a weak SERS active material, can not only increase the SERS ability of the NPG but can also stimulate the long-range effect of the NPG surface to a certain extent, thus increasing the Raman scattering cross section [35].

### 3.4. Detecting Sensitivity of Carbon-Coated NPG and the Calculated Enhancement Factor

The enhancement factor (EF) is often used to evaluate the SERS ability of a substrate. The average enhancement factor was calculated by comparing the Raman spectra of 10^−2^ M crystal violet solution (Figure 5a line Ⅰ) with that of 10^−13^ M crystal violet solution on C@NPG ribbons (Figure 5a line Ⅳ). The integral intensities of characteristic peaks at 442 cm^−1^, 915 cm^−1^ and 1624 cm^−1^ were selected to calculate the average enhancement factor. In the calculation of the enhancement factor of C@NPG, the incident laser diameter is ~5μm. The molecular number of crystal violet detected was calculated by the following formula:N = (N_A_MV_solution_/S_sub_) S_laser_(1)
where N_A_ is the Avogadro constant; M is the molar concentration of crystal violet solution; V_solution_ (10 μL) is the volume of CV solution dropped on the substrate; S_sub_ (~4 mm^2^) is the effective substrate area covered by CV solution; S_larser_ is the cross section area of laser beam (about 5 μm in diameter). The EF was estimated according to the formula
EF = (I_SERS_ ∗ N_NRS_)/(I_NRS_ ∗ N_SERS_)(2)
where I_SERS_ and I_NRS_ refer to the signal intensity of characteristic peaks from CV selected in SERS and NRS (normal Raman scattering, NRS), and N_SERS_ and N_NRS_ are the number of molecules contributing to SERS and NRS signals, respectively [36]. The relevant parameters value and EF are listed in Table 2. The enhancement factors calculated with the three selected characteristic Raman peaks all reach 10^8^. C@NPG could exhibit ultrasensitive SERS detection compared to the previously reported SERS substrates (more details can be found from Appendix A).

In order to further analyze the SERS ability of the substrate, the sensitivity, reproducibility and uniformity of C@NPG have been investigated. Appendix A shows the SERS spectra of CV molecules adsorbed on five different substrates, and it can be seen that the signal intensities are substantially constant, indicating the process of preparing C@NPG is reproducible and attainable. Appendix A shows the Raman spectra of 10^−6^ M CV molecules obtained from one C@NPG substrate at eight different positions, and two characteristic Raman peaks at 439 cm^−1^ and 1172 cm^−1^ are selected for comparison. As shown in the figures, the intensity rarely changed and the calculated standard deviations (RSDs) based on the selected peaks are 6.16% and 6.89%, respectively. Moreover, we tuned the concentration of the CV, and it is possible to detect the signal of 10^−13^ M concentrated CV solution (see Appendix A). Two main Raman peaks at 442 cm^−1^ and 1624 cm^−1^ were traced and analyzed by fitting peak intensity as a function of concentrations so as to estimate the correlation between detection sensitivity and molecule concentration (Figure 5b,c). The fitted results clearly show that the SERS intensity of CV on C@NPG is linearly related to the molecular concentration, with a coefficient of determination of 0.9.

Additionally, C@NPG shows good SERS enhancement for other molecules, such as nicotinamide, which is a water-soluble vitamin, and excessive intake can lead to liver injury. Appendix A shows a detection limit of nicotinamide of 10^−5^ M and an excellent linearity with an R^2^ of ~0.95. Therefore, C@NPG has certain universality and can be applied for sensitive detection of various objects.

## 4. Conclusions

In general, we synthesized C@NPG by one-step dealloying, adopting a pre-annealing AuCu precursor alloy with a carbon material covering. The existence of the surface carbon layer not only avoided the direct contact between the adsorbed molecules and NPG, which can change the adsorption mode of molecules and inhibit photocarbonization, but also improved the electromagnetic enhancement property of the NPG. With optimizing the nanopore basement and carbon layer, it is promising to further improve the optical properties of the hybrid SERS substrate. The method can be used to develop and manufacture more efficient and biocompatible SERS substrates, providing a possible way to realize ultrasensitive sensing with the SERS technique.

## Figures and Tables

**Figure 1 nanomaterials-12-01455-f001:**
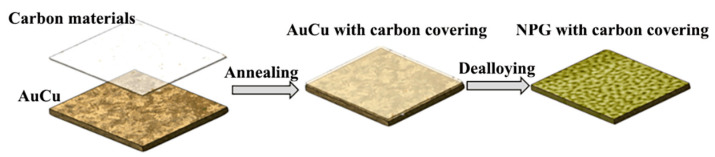
Diagrammatic sketch of carbon-assistant nanoporous gold fabrication process.

**Figure 2 nanomaterials-12-01455-f002:**
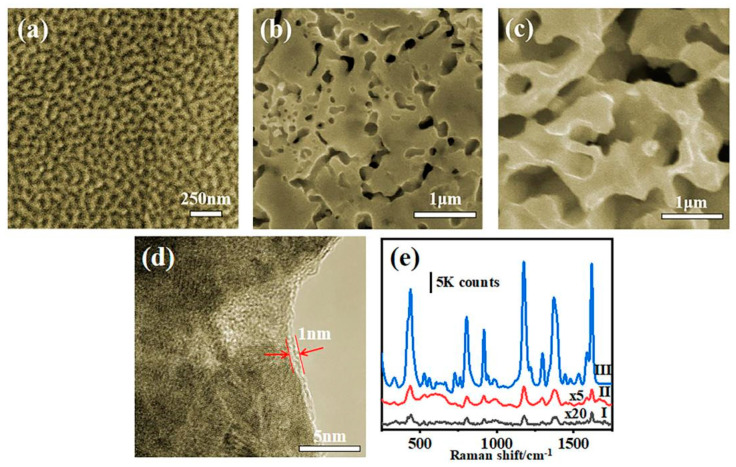
SEM images of NPG prepared with etching precursor alloy (**a**) annealed at 200 °C without carbon covering, (**b**) annealed at 300 °C without carbon covering and (**c**) annealed at 300 °C with carbon covering; (**d**) TEM of C@NPG (annealed at 300 °C with carbon covering), carbon layer is highlighted by red dashed line; (**e**) Raman spectra of 10^−6^ M crystal violet (CV) molecules adsorbed on NPG prepared at different situations (precursor annealed at 200 °C without carbon covering, line I; precursor annealed at 300 °C without carbon covering, line II; precursor annealed at 300 °C with carbon covering, line III). Note: “×5” and “×20” indicate the intensities are magnified by 5 and 20 times for comparison.

**Figure 3 nanomaterials-12-01455-f003:**
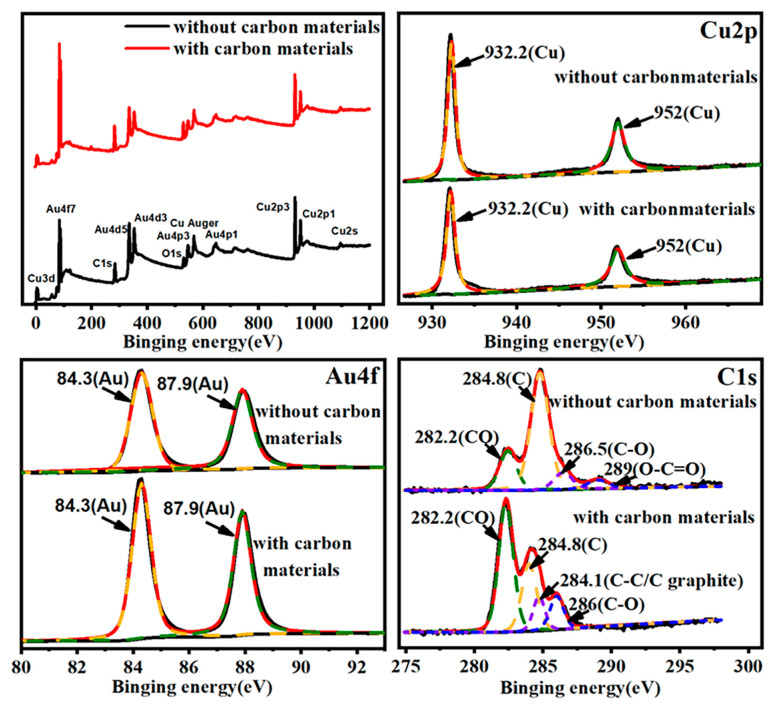
XPS for full survey spectra, Cu 2p, Au 4f and C 1s.

**Figure 4 nanomaterials-12-01455-f004:**
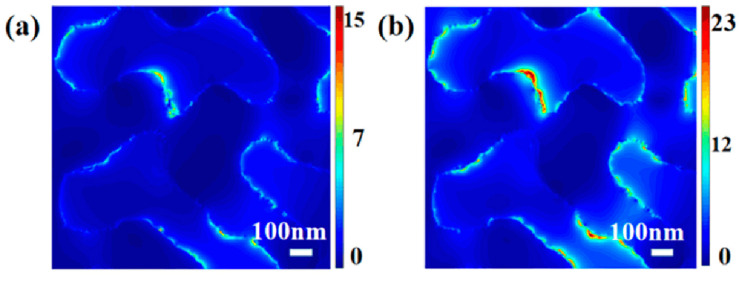
FDTD simulation for local surface electric field distribution of (**a**) bare NPG and (**b**) 1 nm carbon-layer-coated NPG.

**Figure 5 nanomaterials-12-01455-f005:**
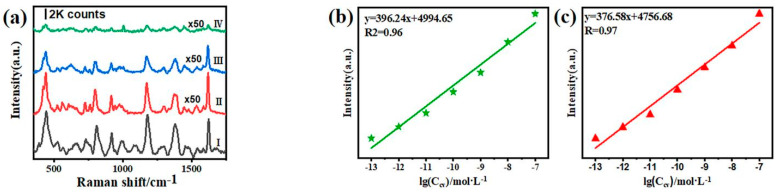
(**a**) Ordinary Raman spectra of CV (I: 10^−2^ M aqueous solution) and SERS spectra of CV molecules with different concentrations on the same SERS substrate (II: 10^−11^ M; III: 10^−12^ M; IV: 10^−13^ M). Note: “×50” indicates that the signal is magnified by 50 times for seeing. SERS intensity as a function of the concentration of the CV molecules: (**b**) 442 cm^−^^1^; (**c**) 1624 cm^−^^1^.

**Table 1 nanomaterials-12-01455-t001:** Comparison and assignments of Raman characteristic peaks between crystal violet aqueous solution and SERS spectrum.

Ordinary Raman Characteristic Peaks of CV Aqueous Solution (cm^−1^)	NPG without Carbon Covering (cm^−1^)	NPG with Carbon Covering (cm^−1^)	Vibration Mode
442	437	439	Ring-C^+^ outside bending vibration
810	804	800	Bending vibration outside ring C-H
1177	1174	1172	Bending vibration in the ring C-H
1379	1378	1371	N-ring Stretching vibration
1624	1620	1618	Ring C=C Stretching vibration bending

**Table 2 nanomaterials-12-01455-t002:** Calculated average enhancement factor with different characteristic Raman peaks.

Raman Peak/cm^−1^	SERS Intensity	NRS Intensity	EF
442	28.68	8409.13	3.4 × 10^8^
919	15.37	4142.82	3.7 × 10^8^
1624	27.99	6892.66	4.06 × 10^8^

## Data Availability

All data concerning this study are contained in the present manuscript or in previous articles, whose references have been provided.

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
