# Peer review of "Carbon-Assistant Nanoporous Gold for Surface-Enhanced Raman Scattering"

_nanomaterials, 2022, doi:10.3390/nano12091455_

Round 1

Reviewer 1 Report

The manuscript submitted by Zhang et al. describes the use of carbon-nanoporous gold SERS substrates on the detection of crystal violet. After carefully evaluating the manuscript, I do not recommend its publication in Nanomaterials.

- The materials and methods section has to be improved. The authors should describe how the substrates and the experiments were carried out in detail. What are the copper and gold starting material? How Au36Cu64 ribbons were prepared? Which carbon materials were used? It was reported the use of HCl 0.1 M to prepare the nanoporous gold substrate. However, figure 2 shows SEM images of the substrates prepared using 0.5 M and 0.1 M.

- Why did the authors choose HCl 0.1M and 0.5M to remove copper?

- The morphology of the substrates reported in Fig.2a, Fig.2b and Fig.2c, cannot be compared bluntly. The experimental conditions used to prepare them are different. To better understand the formation of the substrates, the author should include at least substrates:

  1. With carbon covering at 200ºC-0.5M HCl-24h
  2. Without carbon covering at 300ºC-0.1M HCl-24h
  3. With carbon covering at 300ºC-0.1MHCl-24h

The authors should also report the SERS spectra of CV in all the abovementioned substrates to better understand the effect of the carbon covering and porous formation, size of porous, etc.

- What is the size of the porous and the thickness of the final substrates? Which is the thickness of the carbon materials?

- The authors should include SERS intensity maps.

I recommend the authors test more CV concentrations on the substrates and plot the intensity of the peaks versus concentration of CV to evaluate a possible (or not) linear variation.

- Why the Raman spectra of CV was taken at 10-2M?

- What do the authors mean by: “Obviously, in the presence of carbon materials, the proportion of gold increased, which means more copper was removed (Line 121-122)”

- It is not clear to me what the authors meant by “ the relative intensity of the related peaks between Cu and Au are opposite, indicating that with carbon assistant more copper element was removed from the surface during the dealloying process (Line 156-157). I cannot see that from Figure 3. 

Author Response

Thank you for the comments.We have added and explain.Please see the attachment.

Reviewer 2 Report

This manuscript reports a surface modification method to construct carbon-coated nanoporous gold (C@NPG) SERS substrate.

  1. Materials and Methods section must be revised especially for the details, such as Raman instrumentation and FDTD simulation.
  2. In Figure 2, the image scale difference between (a) and (b) (or (c)) interferes comparison of the nanopore sizes. Also, the authors need to explain the difference of the nanopore size quantitatively.
  3. How thick is the carbon layer on the substrate? Is it a single layer of carbon atoms? The authors need to explain these, because the thickness affects SERS mechanism and because they need to justify their FDTD simulation design.
  4. In the performance of a SERS substrate, not only SERS ability (i.e. high EF) but also signal reproducibility (i.e., low deviation of intensity) is also very important. The authors need to show the deviation of signal intensity of C@NPG substrate.
  5. What are the general applicability of this substrate (extension to detection of many other molecules)?

Please revise the manuscript based on these comments.

Author Response

(The authors gave the same response as above.)

Reviewer 3 Report

The manuscript reports carbon-coated nanoporous gold as an efficient SERS substrate. The manuscript is written well, and the data is of fine quality. However, there are a few points that need to be revised before publication.

Comments: 

  1. The author should provide the stability and reproducibility of the reported SERS substrate.
  2. The author should discuss recently reported nanoporous gold-based SERS substrate in the introduction section. Journal of Alloys and Compounds, 888, 161504 (2021).
  3. In Figure 5, the Author should also demonstrate the SERS spectra of low CV concentrations.
  4. Why did the author use the drop-casting method instead of immersing method?
  5. Discuss the role of chemical mechanisms for SERS enhancement.

Author Response

(The authors gave the same response as above.)

Round 2

Reviewer 1 Report

I appreciate the authors addressing most of my comments. However, I still have some additional observations.

-         The authors should organize the supporting information according to the current form of the manuscript. Also, the figures are mislabelled. For example, the figure representing the “Morphology Characterization and SERS Ability Comparison of NPG with and without Carbon Material for Pretreatment after Dealloying” is labelled as Figure S2 and S1.

-         What are the tiny particles present in the substrate prepared without carbon covering and annealed at 300 ºC (Figure S1b)?.

-         The SEM picture indicating the thickness of the substrate should be included in the manuscript or the supporting information.

-         Regarding my previous comment about the sentence “Obviously, in the presence of carbon materials, the proportion of gold increased, which means more copper was removed (Line 121-122)”. I think, what the authors meant is that the ratio Au:Copper is higher after removing copper for the substrates covered with carbon compared to those not covered.

-         The authors should revise the English spelling and grammar for the whole manuscript.

Reviewer 2 Report

Thank you for your efforts to improve the manuscript.

As only minor revision, I recommend you to revise the unit 'mol/L' to 'M' in the figure S6 and related statements for consistency of the manuscript.

Reviewer 3 Report

The author responded to all the comments. The revised version of the manuscript is now acceptable for publication.
